# An Intelligent System to Improve Diagnostic Support for Oral Squamous Cell Carcinoma

**DOI:** 10.3390/healthcare11192675

**Published:** 2023-10-03

**Authors:** Afonso U. Fonseca, Juliana P. Felix, Hedenir Pinheiro, Gabriel S. Vieira, Ýleris C. Mourão, Juliana C. G. Monteiro, Fabrizzio Soares

**Affiliations:** 1Institute of Informatics, Federal University of Goiás, Goiânia 74690-900, GO, Brazil; julianafelix@discente.ufg.br (J.P.F.); hedenir@discente.ufg.br (H.P.); gabriel.vieira@ifgoiano.edu.br (G.S.V.); fabrizzio@ufg.br (F.S.); 2Federal Institute Goiano, Computer Vision Lab, Urutaí 75790-000, GO, Brazil; 3General Hospital of Goiânia, Goiânia 74110-010, GO, Brazil; hgg.tutoria.fono@idtech.org.br; 4Araújo Jorge Cancer Hospital, Goiânia 74605-070, GO, Brazil; juliana.monteiro@haj.org.br

**Keywords:** oral squamous cell carcinoma, metabolites’ salivary biomarkers, intelligent diagnostic support system

## Abstract

Oral squamous cell carcinoma (OSCC) is one of the most-prevalent cancer types worldwide, and it poses a serious threat to public health due to its high mortality and morbidity rates. OSCC typically has a poor prognosis, significantly reducing the chances of patient survival. Therefore, early detection is crucial to achieving a favorable prognosis by providing prompt treatment and increasing the chances of remission. Salivary biomarkers have been established in numerous studies to be a trustworthy and non-invasive alternative for early cancer detection. In this sense, we propose an intelligent system that utilizes feed-forward artificial neural networks to classify carcinoma with salivary biomarkers extracted from control and OSCC patient samples. We conducted experiments using various salivary biomarkers, ranging from 1 to 51, to train the model, and we achieved excellent results with precision, sensitivity, and specificity values of 98.53%, 96.30%, and 97.56%, respectively. Our system effectively classified the initial cases of OSCC with different amounts of biomarkers, aiding medical professionals in decision-making and providing a more-accurate diagnosis. This could contribute to a higher chance of treatment success and patient survival. Furthermore, the minimalist configuration of our model presents the potential for incorporation into resource-limited devices or environments.

## 1. Introduction

Oral squamous cell carcinoma (OSCC) is among the ten most-prevalent cancer categories worldwide, associated with high mortality and morbidity rates, representing a global public health problem. In 2020, according to statistics from the World Health Organization’s Global Cancer Observatory, lip and oral cavity cancers accounted for more than 377,700 cases worldwide [1]. About 90% of oral malignancies correspond to squamous cell carcinoma [2,3].

OSCC is a malignant tumor that develops from the squamous epithelial cells lining the oral cavity, encompassing the lips, tongue, buccal mucosa, upper and lower gums, and palate, among other areas. Understanding the pathophysiology of OSCC is crucial for early detection, prevention, and effective management of this aggressive cancer. The main risk factors for OSCC include tobacco and alcohol consumption and human *papillomavirus* (HPV) infection present in most patients diagnosed with oral and oropharyngeal cancer, making them critical etiological factors [4]. Moreover, chronic irritation, such as from sharp teeth, ill-fitting dentures, and so forth, and a poor diet in terms of fruits and vegetables are factors. Additionally, socioeconomic factors, such as poor oral hygiene and limited access to healthcare services, also impact early detection and treatment outcomes [5].

The pathophysiology involves a multi-step process of genetic and molecular changes that lead to uncontrolled cell growth and the formation of malignant tumors. As the tumor advances, it can infiltrate nearby tissues, extend to regional lymph nodes, and even metastasize to distant organs. This progression may lead to the temporary or permanent compromise of crucial functions such as speech, voice, swallowing, and mastication and may also be influenced by the chosen treatment approach [6]. In summary, OSCC involves the accumulation of genetic mutations, dysregulation of signaling pathways, invasion, and metastasis. Early detection and intervention are crucial for improving the prognosis and quality of life for individuals with OSCC. Preventive measures, such as avoiding tobacco and alcohol and receiving the HPV vaccine, can also reduce the risk of developing this devastating disease.

Management of all oral cavity cancers must occur in a Multidisciplinary Head and Neck Oncology Team due to the functional and aesthetic implications of the treatment, and the diagnosis of OSCC is traditionally performed by biopsy. OSCC’s 5-year survival rates are around 50%, and most of these patients survive a short time after diagnosis [7]. This scenario is because most tumors are identified late, compromising treatment, prognosis, and patient survival [8]. Given its high mortality, early and accurate diagnosis is extremely important [9].

To better understand the disease and the contribution of early diagnosis, disseminating OSCC signs and symptoms and implementing actions for early detection are essential. In this sense, tools and solutions that help diagnosis and prognosis in the early stages of the disease can be of great value, especially for patients. With early identification or an early prognosis, an increase in survival rates due to the initiation of treatment in the early stages is expected. In addition, a reduction in the mutilating impact of the treatment can also be cited, benefiting the patient and ensuring a better quality of life. These tools help medical professionals decide on the best treatment [10].

Studies indicate that salivary biomarkers can be used for OSCC’s early recognition [11]. Furthermore, salivary collection can be performed in a simple and non-invasive way, and its potential to aid in OSCC diagnosing has been studied by several authors [12,13,14,15,16,17,18].

Similarly, Artificial-Intelligence-guided approaches have shown considerable benefits for early diagnosis and prognosis of oral cancers [19], presenting computational machine learning solutions with improved accuracy of cancer sensitivity, recurrence, and survival predictions [20] and providing a consequent improvement in patient care and survival [21,22,23,24,25].

These approaches have been gaining more and more space in the medical field, with models based on artificial neural networks successfully used in several applications. For example, classification, prognosis, diagnosis, or identification of lung diseases [26,27], chronic kidney diseases [28], macular degeneration and diabetic retinopathy [29], Parkinson’s disease [30], and oral cancer and periodontitis [31,32,33] are among these applications, to name a few.

It is important to note that OSCC is hard to diagnose in the early stages, especially because patients are not aware of the symptoms and signs, which mimic regular oral irritations, ulcerations, leukoplakia, and gingivitis. Therefore, small injuries can be misinterpreted, allowing the cancer to evolve. Classical OSCC diagnosis relies on a biopsy being conducted, required only after clinical observation; however, injuries are already visible and likely too advanced. Therefore, a simple process for collecting salivary samples to gather biomarkers is not only quick, less expensive, and non-invasive, but also can become a general clinical protocol for prevention. Finally, analyzing the biomarkers is a hard task, so our study aimed to evaluate their impact on classifiers for the early diagnosis of OSCC. Therefore, this work aimed to develop a method with a shallow artificial neural network to distinguish data from people with OSCC and healthy people from salivary biomarker data. The proposal was evaluated using the cross-validation and data augmentation strategies, and the calculated accuracy, sensitivity, and specificity metrics, combined with the simplicity and portability of the proposed method, point to a way to use the proposal in practice. Some of our main contributions were:Evaluating the use and performance of a feed-forward shallow artificial neural network in the classification of control and OSCC cases through salivary biomarkers;Evaluating the impact of using data augmentation to class balancing in the model building;Classifying salivary biomarkers according to their importance values based on the quadratic weights assigned by a support vector machine (SVM) classifier;Selecting a minimum set of salivary biomarkers to preserve high precision for the classification task.

This work is organized as follows: First, the related work is presented in Section 2, and Section 3 presents the materials and methods used in this work, including information from the database and the characteristics of the implemented neural network. Next, a comparative analysis and discussion of the results obtained are presented in Section 4. Finally, our conclusions and future work follow in Section 5.

## 2. Related Work

Artificial Intelligence has been successfully used to assist in the diagnosis and prognosis of OSCC cases by analyzing hyperspectral clinical, pathological, and radiographic images. Halicek et al. [34,35] used hyperspectral imaging and convolutional neural networks to diagnose aerodigestive tract tumors with a better sensitivity of 81%. Ariji et al. [36] used computerized tomography (CT) images to detect cervical lymph node metastasis and infer disease-free survival. Fujima et al. [37] similarly used F-fluorodeoxyglucose (FDG) uptake by positron emission tomography (PET/CT) and achieved 80% accuracy in predicting survival. Pathological images were used by Folmsbee et al. [38] to identify oral cavity cancerous tissues among six other tissue classes with an accuracy of 96.37%, while Das et al. [39] classified biopsy images with similar performance to the pathological classification system in [40], achieving 97.51% accuracy.

Although the importance of clinical images in treating OSCC and of Artificial Intelligence solutions to aid in the diagnosis and prognosis of this disease is undeniable, image acquisition is not trivial, requiring highly complex human and material resources, while Artificial Intelligence applications that model the training steps need proper data preparation and labeling in an expensive and time-consuming process. Therefore, we must consider other data types such as gene expression data, spectral data, autofluorescence, and saliva metabolites.

Thus, specifically concerning chemical methods, salivary biomarkers are an excellent alternative for analyzing and monitoring OSCC cases. Salivary biomarkers offer good accuracy in the early recognition of this disease. In addition, collecting salivary samples is a simple, non-invasive process with lower costs than clinical imaging.

Hu et al. [18] demonstrated that protein biomarkers of the human salivary proteome have a high capacity to discriminate OSCC patients from healthy patients, indicating that the combination of some salivary proteins reaches a sensitivity of up to 90% in this task and showing a promising approach for the search for OSCC biomarkers.

Using autofluorescence spectroscopy, Yuvaraj et al. [15] performed a pilot study demonstrating marked differences in the spectral signatures between healthy and OSCC patients, obtaining 85.7% sensitivity when using the fluorescence emission spectrum at a 405nm excitation. The authors further suggested that the reasons are linked to endogenous porphyrin, nicotinamide adenine dinucleotide (phosphate), reduced (NAD(P)H), and flavin adenine dinucleotide (FAD) in exfoliated saliva cells.

Sridharan et al. [41] used quadrupole time-of-flight (Q-TOF) liquid chromatography–mass spectrometry to assess the metabolomic profile of OSCC, oral leukoplakia (OLK), and control groups. The authors found a significant increase in 1-methylhistidine, inositol 1,3,4-triphosphate, d-glycerate-2-phosphate, 4-nitroquinoline-1-oxide, 2-oxoarginine, norcocaine nitroxide, sphinganine-1-phosphate, and pseudouridine in OLK and OSCC in comparison to control groups.

Song et al. [12] used conductive polymer spray–mass spectrometry (CPSI–MS) to quantify saliva samples from OSCC, patients with premalignant lesions, and healthy individuals. The authors showed that using CPSI–MS with machine learning (ML) can be a viable tool to distinguish OSCC and premalignant lesions from healthy conditions in real-time, with an achieved accuracy of 86.7%.

Ishikawa et al. [42] developed a multiple logistic regression model to differentiate OSCC from oral lichen planus. They realized a comprehensive analysis of fourteen hydrophilic metabolites in the saliva samples by capillary electrophoresis–mass spectrometry, showing high discrimination of the combination of indole-3-acetate and ethanolamine phosphate metabolites, resulting in 85.6% of the area under the curve (AUC).

Recently, Kouznetsova et al. [32] studied metabolic pathways related to oral cancer and periodontitis and, using a deep learning neural network, achieved the best accuracy of 79.54% in the classification of these cases.

Finally, Costa et al. [43] analyzed saliva samples using gas chromatography–mass spectrometry with a data-mining approach to find the best biomarkers to diagnose OSCC. Using feature selection classification and random forest classification algorithms, they reported an AUC of 91% with only three metabolites versus 76% when using all 51 metabolites analyzed.

As we have seen, OSCC can be approached differently from gene expression data, spectral data, autofluorescence, salivary metabolites, cytology images, CT images, and clinicopathological images [33]. Thus, some studies that used clinical images and salivary biomarkers were described to illustrate the OSCC research scenario, which, despite being diversified, still has limitations in chemical and computational advances. Therefore, to overcome some of the limitations of this field and contribute to the research, we combined Artificial Intelligence techniques to deliver a low-cost computational model and real-time response. Our proposal uses the same base of salivary biomarkers used by [43] and presents accuracy, sensitivity, and specificity values superior to previous works for several validation schemes analyzed.

## 3. Material and Methods

We developed a minimalist and effective oral squamous cell carcinoma diagnosis support method. Our proposal consists of two more significant cohesive steps, structuring a pipeline that begins with the biomarker processing and, then, follows with artificial neural network construction. Figure 1 provides an overview of the proposed method, described in the following subsections.

### 3.1. Biomarker Salivary Database

The dataset used in this study was the same used by and publicly available from [43]. The research volunteers were adults over 18 years old, treated at the Oral Medicine Outpatient Clinic of the Department of Biosciences and Oral Diagnosis of the Institute of Science and Technology of São José dos Campos (ICT-UNESP). All volunteers signed an informed consent form (ICF) to participate in the research approved by the ethics committee under Protocol Number 1.033.312/2015 PH/CEP.

The database comprises data from chemical analyses of salivary samples obtained from 68 people, of which 27 were in the group with a positive diagnosis for OSCC and 41 people comprised the control group. Among the two groups of people, information on smokers, ex-smokers, and non-smokers was also collected and available. Table 1 presents a summarized description of the demographic and smoking profile of the 68 study participants of this database.

For the composition of the group of individuals with OSCC, age over 18 years and concomitant diagnosis of OSCC were considered as the inclusion criteria, disregarding patients who had already undergone oncological treatment, such as surgery, radiotherapy, or chemotherapy for any type of cancer, including oral cancer. For patients in the control group, the criterion was age over 18 years, without a diagnosis of malignant neoplasm in life, and volunteering for research.

From each participant, a salivary sample of 300 μL was collected. The sample was mixed with 300 μL of ethanol, shaken for 30 s, and then, centrifuged. Over each salivary sample, 51 metabolic biomarkers were extracted from gas chromatography–mass spectrometry.

The biomarkers were as follows: Acetoacetic.acid, Adenine, Arabitol, Batyl.alcohol, Cadaverine, Caproic.acid, Citramalic.acid, Cysteine, Dopamine, Eicosapentaenoic.acid, Elai-dic.acid, Galactosamine, Galactose, Glucono.1.5.lactone, Glucosamine, Glucose, Glucuronic.acid, Glutamic.acid, Glycine, Glycolic.acid, Hydroxylamine, Inositol, Isoleucine, Linoleic.acid, Maleic.acid, Mannitol, Mannose, Margaric.acid, Meso.Erythritol, Methionine.sulfone, Norepinephrine, Octadecanol, Oleic.acid, O.Phosphoethanolamine, Ornithi-ne, Palmitic.acid, Palmitoleic.acid, Phenylalanine, Putrescine, Ribitol, Sarcosine, Threo-nic.acid, Threonine, Tryptamine, Tyramine, Uridine, X2.Aminoethanol, X2.Deoxy.gluco-se, X2.Hydroxyisovaleric. acid, X3.Aminoglutaric.acid, X3.Aminopropanoic.acid.

### 3.2. Data Augmentation

As described in Section 3.1, our original database (Orig-DB) contained 68 samples of control and OSCC cases, with more prevalence of samples of control cases (41) in contrast to OSCC cases (27). Thus, to overcome this imbalance between these classes, we also evaluated a more-extensive dataset in which the OSCC minority class was synthetically increased by the synthetic minority oversampling technique (SMOTE) to promote the balance between the classes. As a result, in this new dataset (Aug-DB), the total number of samples was 82, with an equal number of control and OSCC cases.

SMOTE is a data-augmentation technique that performs resampling by taking the sample neighborhood as a reference and creating a new instance connecting to this neighborhood. He et al. [44] gave the following definition for this technique:

“SMOTE first selects a minority class instance ‘a’ at random and finds its ‘k’ nearest minority class neighbors. The synthetic instance is then created by choosing one of the ‘k’ nearest neighbors ‘b’ at random and connecting ‘a’ and ‘b’ to form a line segment in the feature space. The synthetic instances are generated as a convex combination of the two chosen instances ‘a’ and ‘b’.”

When generating our synthetic samples, we used a value of *k* = 5 in SMOTE. This value was defined empirically, considering the number of original samples available (27), in order to balance the important information and have better sample diversity. Thus, we sought to improve the generalization of the model, avoiding both overfitting and underfitting.

To understand the effect of the resampling by neighborhood performed by the SMOTE technique, in Figure 2, we provide the plots of the t-distributed stochastic neighbor embedding (t-SNE) [45] of the original Biomarker Salivary Database and after the data augmentation by SMOTE was performed.

These t-SNE plots were generated from normalized data by the Z-score using the well-known Euclidean distance [46] and show the sample distribution in the Orig-DB and in the Aug-DB, highlighting how the new synthetic samples generated by SMOTE were positioned. These plots also help explain the reason for evaluating the balance effect between classes, taking the similarity of the samples between the classes. Furthermore, for better understanding, we invite the reader to see the original work by Nitesh Chawla et al. [47] with a detailed description of the SMOTE technique.

### 3.3. Ranking and Feature Selection

Feature selection aims to reduce the number of input variables in a learning model and optimize model performance using those with the most-significant relevance. Some benefits include less model development and training time, more-efficient use of computational resources, and the avoidance of model overfitting. In addition, keeping irrelevant features can also result in models with low generalization power.

In this work, the SVMAttributeEval algorithm was used as a solution for this task, which evaluates the value of a feature using an SVM classifier, which then classifies it by the square of the weight assigned by the SVM. If we have a multiple-class model, the algorithm ranks each class’s features separately using a one-vs.-all (OvA) strategy and, then, “negotiating” from the top of each stack to obtain a final ranking. A more-detailed explanation of this algorithm is given in [48]. The algorithm parameter values that we used in this work were as follows:Constant elimination threshold = 0;Filtering used by SVM = normalization of training data;Tolerance *T* = $1.0^{-10}$;Complexity *C* = 1.0;Epsilon *P* = $1.0^{-25}$;Constant rate of feature elimination = 0;Feature elimination percentage rate = 0;Evaluator check = yes.

Notably, in both the Orig-DB and the one Aug-DB by SMOTE, we used the *SVMAttributeEval* algorithm. Table 2 shows the ranking position of each biomarker in the databases. Figure 3 shows the violin plot (normalized values by the Z-score technique) of the 15 highest-ranked features from the Orig-DB, while Figure 4 is their correlation heatmap, where darker colors indicate less dependency between features.

### 3.4. Artificial Neural Network

The success of neural networks in different fields of knowledge has placed them as the first option for solving several [49] problems. However, despite their success, most neural networks, especially the so-called deep neural networks, present characteristics of requiring equipment with high computational power, the need for a large volume of data, and much time for training. Thus, these characteristics make their use in solutions designed for environments and equipment with limited resources unfeasible. In this sense, feed-forward neural networks (FFNNs), the first and simplest category of artificial neural networks developed [50], represent a good alternative because they present excellent performance and efficiency in balance with the required resources. In these networks, connections between nodes do not form a loop, so information only moves in one direction—forwards—from input layer nodes, through hidden layer nodes, to output data layer nodes [51].

In this study, we chose these networks for their great predictive capacity, independent of the probability distribution information of the data. Furthermore, these networks have been successfully applied in a wide range of medical applications, including Alzheimer’s disease [52], chronic kidney disease [28], lung cancer detection [53], and the prediction of outbreaks of COVID-19 [54,55].

The parallel structure and the ability to improve performance through experience explain the efficiency of these networks. Therefore, we built a shallow FFNN to support diagnosing oral squamous cell carcinoma cases among normal (healthy/control) cases. The proposed network was built with a minimalist structure, delivering good accuracy and requiring as few computational resources as possible. It consists of fully connected layers. At the network entrance, a set of biomarkers is passed through two hidden layers, each with 3 neurons, and then arrives at the output layer, which determines two possible classes: cancer and control. The schematic of the constructed network is shown in Figure 1.

Other network parameters include the Levenberg–Marquardt backpropagation training function [56,57], which updates the weights (w) and bias (b); the Softmax transfer function in the output layer nodes; a maximum number of epochs equal to 50; and a maximum number of validation failures equal to 5 to prevent overfitting. Other model optimization parameters include the mean-squared error (MSE) loss function with the regularization rate equal to $1\times 10^{-5}$, the hyperbolic tangent sigmoid (Tansig) on the hidden layer nodes to produce faster output rates [58], and cross-validation for the batch size. Table 3 summarizes the optimizing parameters of our models.

### 3.5. Data Splitting and Model Validation

In a neural network performance validation, a model is usually trained with available data, called a training set. Then, the model’s inference power is evaluated using newly collected data or a set of previously separated data unseen by the model. The latter is called a test set.

In this study, we used the cross-validation (CV) technique to measure the generalization power of the proposed model. This technique provides a good measure of generalization by splitting samples from a dataset into mutually exclusive subsets, which are then used in the training and testing rounds. Cross-validation, thus, assesses how the results of a statistical analysis of the model generalize to an independent dataset. In problems where the modeling objective is prediction, this technique is widely used [59].

In the validation process of this study, two different cross-validation schemes were used, the first scheme with a dataset divided into *k* exclusive subsets—*k*-fold cross-validation (*k* = 10)—with nine subsets being used for training and the other used for testing, in a process that was repeated ten times, alternating each time the training and test sets. In the second scheme, a particular case of k-fold cross-validation called leave-one-out cross-validation (LOOCV), we divided the dataset by the number of available samples (*k* = *n*). Thus, considering *n* available samples, each evaluation run uses a training set with n−1 samples, and the sample left out is used for testing, repeating the process *n* times. For both schemes, the reported results always correspond to the average obtained in each test case.

### 3.6. Evaluation Metrics

Public databases related to OSCC are rare in the literature, making it difficult to apply more-current techniques, such as deep learning, which require a large volume of data. Therefore, comparing these more-current and classic techniques becomes a challenge in evaluating the significance and weight of the results. Thus, the cross-validation schemes used in this study and described in Section 3.5 are viable and valuable alternatives for evaluating the results.

In this study, we considered the following evaluation metrics: *accuracy* (*ACC*), which represents the rate of correct results among the total number of cases examined; *sensitivity* (*TPR*), which represents the rate of individuals belonging to the positive class (individuals with OSCC) who were correctly identified as OSCC, and *specificity* (*TNR*), which measures the percentage of people belonging to the negative class (healthy control patients) who were correctly identified.

Thus, after the completion of all test cases for both cross-validation schemes, the average results of the accuracy, sensitivity, and specificity were calculated as follows:(1)$$\begin{array}{cc} Accuracy\;(ACC) & =\frac{TP+TN}{TP+TN+FP+FN} \end{array}$$(2)$$\begin{array}{cc} Sensitivity\;(TPR) & =\frac{TP}{TP+FN} \end{array}$$(3)$$\begin{array}{cc} Specificity\;(TNR) & =\frac{TN}{TN+FP} \end{array}$$
where true positive (TP) is the number of OSCC cases correctly identified as such, true negative (TN) is the number of correctly identified control cases, false positive (FP) is the number of control cases incorrectly labeled as OSCC cases, and false negative (FN) is the number of OSCC cases that were labeled as control cases.

Other metrics considered in this study included the confusion matrix, ROC curves, regression analysis, and area under the ROC curve (AUC).

### 3.7. Setup Configuration

The experiments were carried out on a desktop computer with a Core^TM^ i5-7500 CPU @3.4GHz, 8 GB of RAM, Windows 10—64-bit, without GPU acceleration. We used the Matlab R2014a program with the packages Deep Learning Toolbox and Signal Processing Toolbox installed to build the network. The other settings adopted were the environment’s default.

## 4. Discussion and Analysis of Results

### 4.1. Feature Ablation

To define the minimum amount of biomarkers/features needed to build intelligent artificial neural network models to assist in OSCC diagnosis, we first used the *SVMAttributeEval* algorithm both in the Orig-DB and in the one Aug-DB by SMOTE, ranking the biomarkers as described in Section 3.3.

Thus, after ranking the features, we evaluated the model’s accuracy in each validation scheme, starting from a subset with only the first best-ranked feature for incremental subsets to the subset with all 51 biomarkers.

Figure 5 and Figure 6 show, respectively, the model accuracy achieved for each feature subset under the 10-CV and LOOCV validation schemes in the Orig-DB and Aug-DB databases, from which we can arrive at the following observations:

(i)The classification of features by the SVMAttributeEval algorithm does not guarantee better accuracy for models built with subsets with a more-significant number of features, and the oscillations of the accuracy values, more evident in the 10-CV validation scheme of Figure 5, may be related to the fact that this ranking considers the importance of the characteristics for an SVM classifier and not specifically for the FFNN used in the models;(ii)The balancing of cancer and control classes in the Aug-DB by the SMOTE resampling technique does not result in an improvement in accuracy and, when using the 10-CV validation scheme, requires a more-significant number of features to achieve similar accuracy to the Orig-DB with unbalanced classes, while for the LOOCV validation scheme, the opposite occurs;(iii)The best performance, least number of features, and highest accuracy were obtained by the LOOCV scheme on the Aug-DB by SMOTE (Figure 6b), explained by the lower selection bias, favored even more by the balance of classes and by the lower tendency to overestimate test errors, as it was trained n − 1 times;(iv)Finally, the FFNN models were efficient for all schemes, reaching expressive values of accuracy with a reduced number of features (above 90% with less than ten features).

Furthermore, with regard to the validation schemes, although the reduced dataset size and performance justify the use of the LOOCV scheme, we must keep in mind that it is subject to greater variance and overfitting, high computational cost, and reduced interpretability. Therefore, we reinforce that the choice of the best training scheme must carefully consider the advantages and disadvantages and the suitability for the task to be performed.

That being stated, after evaluating the models for each training scheme and dataset and considering the different feature subsets, we selected the feature subset with the best accuracy result in each configuration. This means 14 and 27 features for the Orig-DB and 20 and 14 features for the Aug-DB, respectively, within the context of the 10-CV and LOOCV schemes. In the following sections of this work, we present additional results from each of these configurations.

### 4.2. Performance Model for Different Validation Schemes

Two different cross-validation schemes were used to assess the model’s generalization, as described in Section 3.5. These schemes were tested on the Orig-DB and Aug-DB by SMOTE. This evaluation of the different validation schemes and datasets aimed to point out the impact of their choices and the advantages or not of adopting one option over another, as already initially discussed in Section 4.1.

Accordingly, we now report more-detailed results for each of these configurations, considering only the subset of biomarkers/traits that performed better in the evaluated configuration (cross-validation scheme and dataset).

In Figure 7, we present the confusion matrices for each evaluated configuration, including the cross-validation schemes (10-CV and LOOCV) and the databases (Orig-DB and Aug-DB). These matrices illustrate that the model maintained stable performance across different configurations, consistently achieving accuracy above 98.5% and a sensitivity (TPR) of at least 96.3%.

With regard to the impact of the sample balancing promoted by SMOTE, it led to an improvement in the TPR (from 96.30% to 100% in the 10-CV scheme) and can reduce the number of features required for the model to achieve its best accuracy (from 20 to 14 in the LOOCV scheme). In terms of cross-validation schemes, the LOOCV technique demonstrated superior accuracy performance compared to 10-CV, albeit with the trade-off of longer training times, which becomes impractical for larger datasets.

Some of these observations can be attributed to the imbalance ratio of 1:1.5 (27:41) in our original database and the utilization of the SMOTE approach. As discussed in [60], the SMOTE approach mitigates the variance of the minority class, leading to an improved TPR, but with a relatively modest impact on the overall classifier performance.

Thus, in light of these findings, we can conclude that the choice of the cross-validation scheme and the use of the data augmentation in our study had minimal influence on the results of the confusion matrices. However, other factors may be considered, as we will see below.

The ROC curves, displayed in Figure 8, illustrate the model’s diagnostic capability as the discriminant threshold varies. Notably, the ROC curves exhibited a consistent behavior, echoing the confusion matrices’ results. For instance, in the 10-CV scheme, we observed a higher AUC value of 0.993 on the Aug-DB (Figure 8c) compared to the 0.965 value on the Orig-DB (Figure 8a), while for the LOOCV scheme, there was no difference in the value of the AUC for the database used.

These ROC curves are of fundamental importance in the interpretation of the results, since their resilience against the potential influence of the class distribution and balance makes them robust indicators of the performance of our model. Furthermore, they have a direct and intrinsic connection with the cost–benefit analysis, a fundamental aspect of diagnostic decision-making.

In this regard, the consistency demonstrated by the curves in Figure 8 with optimal operating points close to 1 across the various configurations we assessed serves as compelling evidence of our model’s exceptional discriminative power in the presented context.

Figure 9 brings a view of the model’s linear regression analysis. Scatter plots between the actual and predicted classes show a strong model correlation coefficient, indicating that the model prediction accurately follows the actual class and achieves a correlation coefficient of at least 0.953 (Figure 9a). Another highlight in these plots is the fit line, which practically overlaps the reference line (Y=T) for all configurations, pointing to a model with an excellent fit to the data, regardless of the validation scheme or dataset used.

Figure 10 presents sample error curves of the training process for 10-CV and LOOCV with the original data (Orig-DB) and balanced data with SMOTE (Aug-DB). The training was set up to stop when the MSE increased or reached 50 epochs. As can be seen, models trained without augmentation stopped at 50 epochs, reaching errors of $3.1298\times 10^{-8}$ and $2.6377\times 10^{-5}$ for 10-CV and LOOCV, respectively, while models trained with data augmentation stopped at 21 and 26 epochs, reaching errors of $1.7284\times 10^{-12}$ and $3.7319\times 10^{-13}$, for 10-CV and LOOCV, respectively. Therefore, the data augmentation was a great improvement in terms of training, since significantly fewer MSEs were reached in fewer epochs, reducing the risk of overfitting.

Finally, Figure 11 shows that the gradient, which represents the slope of the tangent of the graph of the function or simply the network error rate, converged more quickly in the Aug-DB, as well as presenting a lower μ (mean error), which represents the control parameter for the backpropagation neural network that we modeled. Finally, the validation check reflected the error in each cycle pass and was zero for all schemes, indicating good network performance, as seen in the figures.

### 4.3. Comparison with the Literature

Table 4 presents a brief quantitative comparison considering our proposal and some literature studies on diagnosing OSCC cases. Although some works have used different configurations (cross-validation scheme and database) and experimental evaluations, they also have common characteristics, such as measuring the AUC performance and using salivary biomarkers. This comparison is interesting to position our work concerning peers and reveals the principal methodologies used in approaching the diagnosis of OSCC cases. In this sense, our proposal presents, in general terms, better results than the compared studies, both for those that used the same database and those that used other databases. For example, in Costa et al. [43] and Fonseca et al. [61], the same public database was used in a 10-CV validation scheme, while Song et al. [12] used another public database but on a 20-CV validation scheme. The other studies used private databases combining LOOCV validation schemes or statistical analyses.

Thus, in contrast to the compared studies, our proposal was evaluated under two different validation schemes while analyzing the impact of data balancing by SMOTE resampling. In addition, our proposal still ranked the biomarkers by taking the weights of a classified SVM and evaluated the smallest subset necessary to achieve the best performance and accuracy of the model, reaching a minimum ACC and AUC of 98.53% and 96.48% respectively, when using the unbalanced Orig-DB (10-CV) validation scheme.

Finally, we highlight the computational effort required for our model proposed here. The total time spent to perform all model training in the 10-CV validation scheme was only 1.07 s on both the Orig-DB and Aug-DB, while in the LOOCV validation scheme, these times were 5.7 s and 8.2 s, respectively. Furthermore, the average inference time of a sample for all configurations was approximately $2.85\times 10^{-4}$ s, and the amount of memory required for the entire model was approximately 6 MB for each analyzed configuration. These characteristics make it possible to use our proposal, regardless of the configuration adopted (database and validation scheme), in real-time solutions, environments, and equipment with limited resources.

## 5. Conclusions

This study presented a shallow neural network model of the FFNN-type with low computational cost to aid in diagnosing OSCC cases. Using 51 biomarkers extracted from salivary samples from people diagnosed with OSCC and healthy people, we showed that we can effectively contribute to the early detection of OSCC cases with different numbers of these biomarkers. The analysis of a biomarker-selection process by the SVMAttributeEval algorithm was presented for four different configurations, combining two cross-validation schemes (10-CV and LOOCV) applied to an original dataset (Orig-DB) and an augmented dataset by the SMOTE technique (Aug-DB), which was used for balancing between classes. The feature-selection process yielded different subsets for each configuration, being 14 and 27 features for the Orig-DB and 20 and 14 features for the Aug-DB, respectively, within the context of 10-CV and LOOCV. The results for all configurations showed high values of the model’s precision, sensitivity, and specificity. The proposal significantly surpassed previous studies in the literature both for those using the same database and those using other databases.

As a novelty, our model combines the low use of computational resources with high accuracy for the early diagnosis of OSCC with different numbers of salivary biomarkers. Furthermore, the validation in different configurations indicated that our model design, with fewer degrees of freedom, was effectively less prone to bias and overfitting. Thus, the minimalist characteristics make our proposal a viable and inexpensive solution for embedded systems, environments, and equipment with limited processing and storage capacity. Hence, it can effectively aid in early OSCC case detection and be used as a tool in the active search for OSCC cases in health control programs.

In this sense, we concluded that the use of salivary biomarkers combined with a shallow neural network model presents a great potential to assist in the decision-making of medical professionals, providing a more-assertive and early prognosis of OSCC cases and, with this, contributes to a better chance of cure and patient survival. As future advances, we point out tests in other bases of salivary biomarkers and the identification of the stage of the disease. Furthermore, different optimization parameters can be experimented with, as well as other validation schemes and feature-selection techniques.

## Figures and Tables

**Figure 1 healthcare-11-02675-f001:**
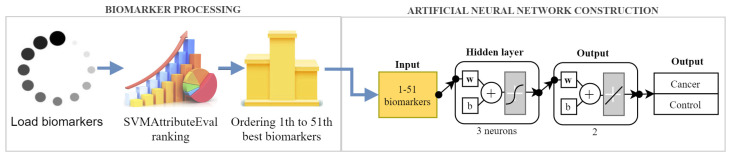
Overview of the proposed method. The diagram is divided into two main steps. In the first step, the biomarkers are classified using the SVMattributeEVal algorithm. In the second step, different numbers of these biomarkers are utilized to build an optimized artificial neural network for the diagnosis of OSCC.

**Figure 2 healthcare-11-02675-f002:**
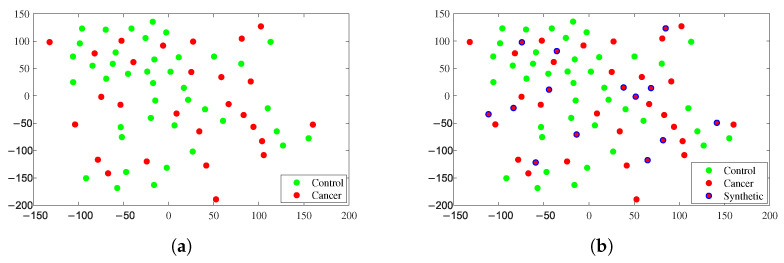
t-SNE plots showing the distribution and dispersion of samples between classes used for OSCC diagnosis. In (**a**), the resulting t-SNE plot with Euclidean distance over the original data distribution samples (Orig-DB). In (**b**), the resulting t-SNE plot with Euclidean distance over the SMOTE-augmented synthetic data samples (Aug-DB). Synthetic samples were obtained by SMOTE (*k* = 5) from the original samples and are highlighted with a blue border.

**Figure 3 healthcare-11-02675-f003:**
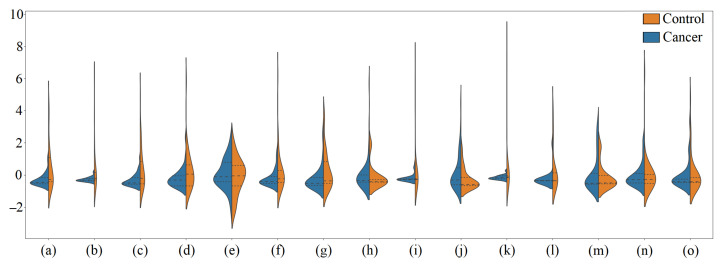
Violin plot illustrating the normalized distribution (by Z-score) of the values of the top 15 features in each class: (**a**) X2.Hydroxyisovaleric.acid, (**b**) O.Phosphoethanolamine, (**c**) Batyl.alcohol, (**d**) Threonic.acid, (**e**) Margaric.acid, (**f**) X2.Deoxy.glucose, (**g**) Threonine, (**h**) Adenine, (**i**) Tyramine, (**j**) Maleic.acid, (**k**) Tryptamine, (**l**) Putrescine, (**m**) Norepinephrine, (**n**) Hydroxylamine, (**o**) Dopamine.

**Figure 4 healthcare-11-02675-f004:**
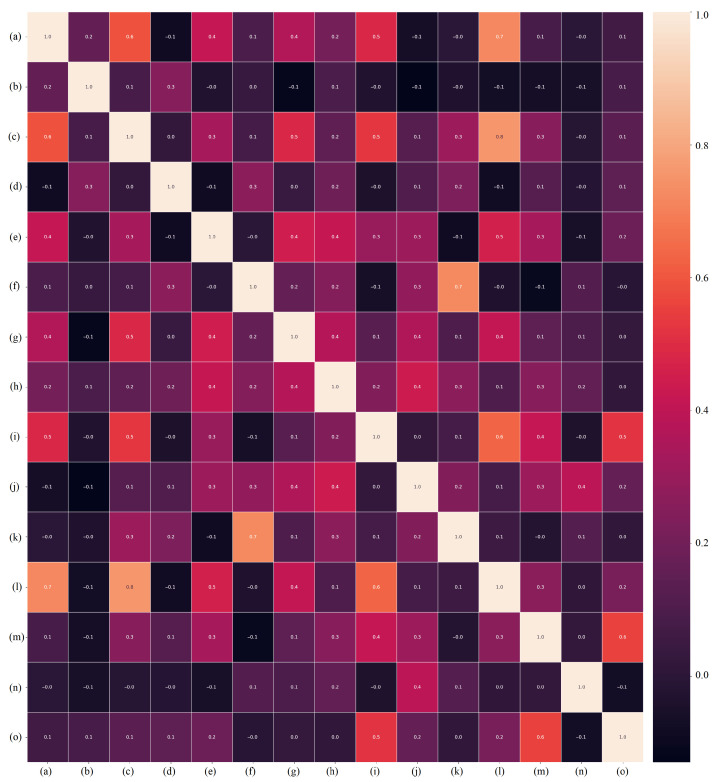
The correlation heatmap between the 15 highest-ranked biomarkers: (**a**) X2.Hydroxyisovaleric.acid, (**b**) O.Phosphoethanolamine, (**c**) Batyl.alcohol, (**d**) Threonic.acid, (**e**) Margaric.acid, (**f**) X2.Deoxy.glucose, (**g**) Threonine, (**h**) Adenine, (**i**) Tyramine, (**j**) Maleic.acid, (**k**) Tryptamine, (**l**) Putrescine, (**m**) Norepinephrine, (**n**) Hydroxylamine, (**o**) Dopamine. Light colors indicate high correlation, while darker colors indicate less dependence.

**Figure 5 healthcare-11-02675-f005:**
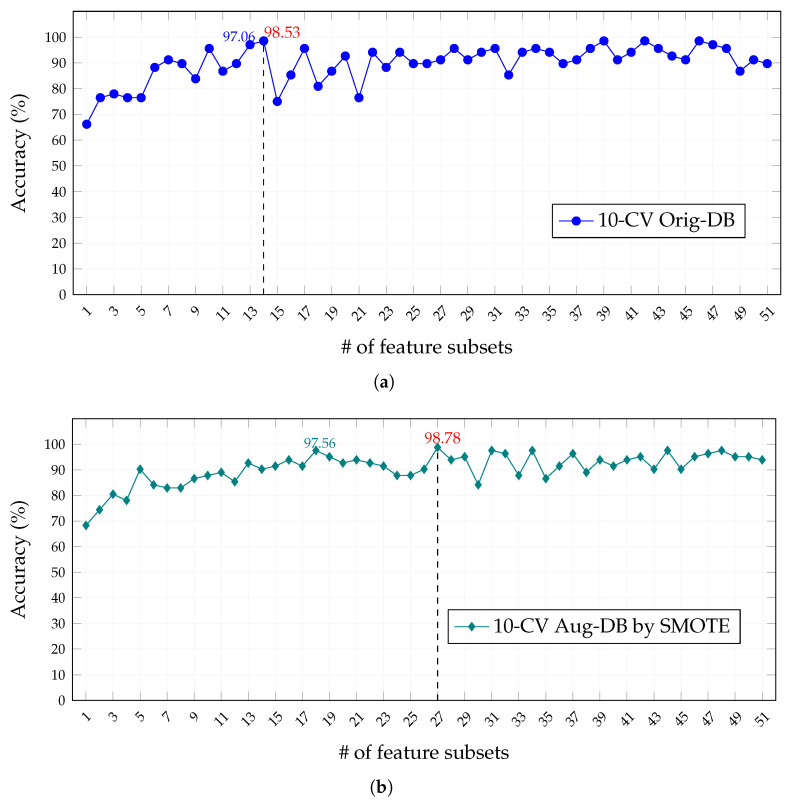
Accuracy performance as a function of selected features for the 10-fold cross-validation (10-CV) scheme in (**a**) Results of classification by 10-CV with samples from Orig-DB and in (**b**) Results of classification by 10-CV with samples from Aug-DB. Red values indicate the best accuracy achieved.

**Figure 6 healthcare-11-02675-f006:**
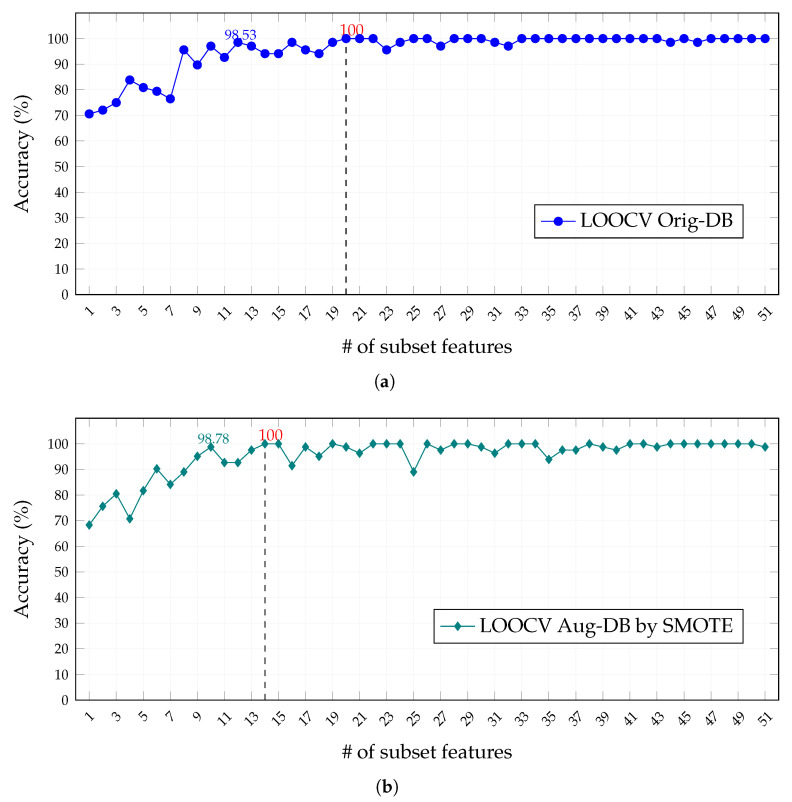
Accuracy performance as a function of selected features for the leave-one-out (LOOCV) scheme in (**a**) LOOCV classification results with Orig-DB samples and in (**b**) LOOCV classification results with Aug-DB samples. Red values indicate the best accuracy achieved.

**Figure 7 healthcare-11-02675-f007:**
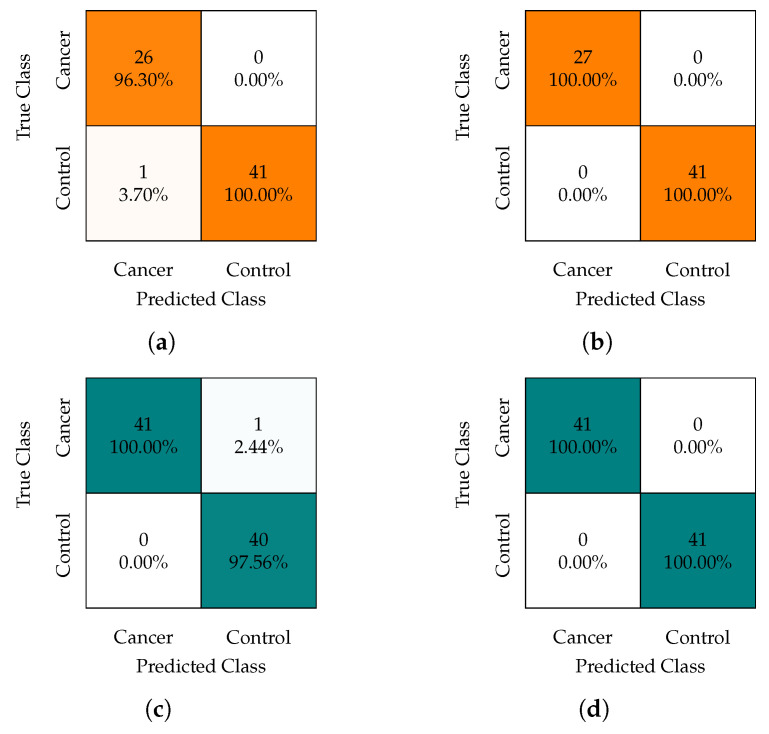
Confusion matrices obtained from the best result achieved by our artificial neural network model in the 10-fold cross-validation (10-CV) and leave-one-out cross-validation (LOOCV) schemes. Subfigures (**a**,**b**) of the original database (Orig-DB) in the 10-CV and LOOCV schemes, with 14 and 20 features, respectively. Subfigures (**c**,**d**) of the SMOTE-augmented database (Aug-DB) in the 10-CV and LOOCV schemes with 27 and 14 features, respectively.

**Figure 8 healthcare-11-02675-f008:**
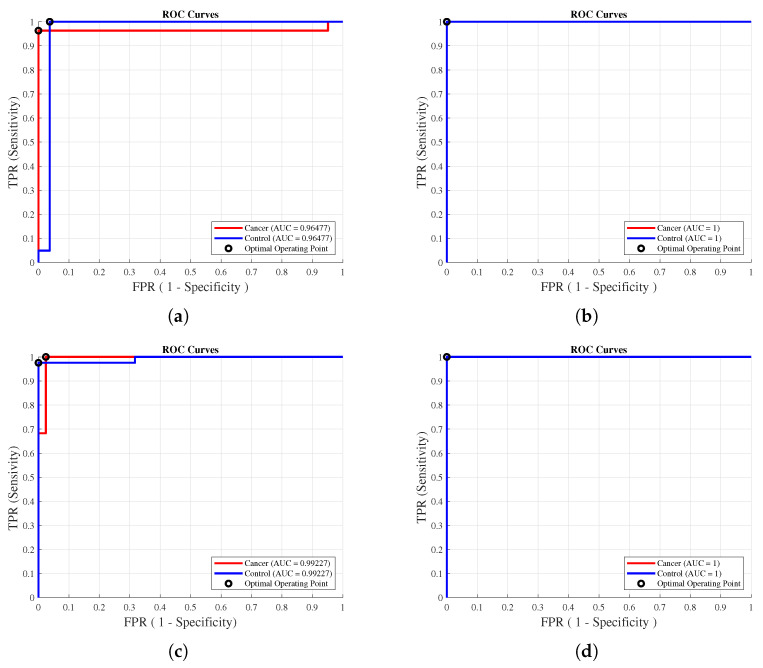
ROC curves of the model with the best subset of features in each training scheme. Subfigures (**a**,**b**) of the original database (Orig-DB) in 10-CV and LOOCV schemes, with 14 and 20 features, respectively. Subfigures (**c**,**d**) of the SMOTE-augmented database (Aug-DB) in the 10-CV and LOOCV schemes, with 27 and 14 features, respectively.

**Figure 9 healthcare-11-02675-f009:**
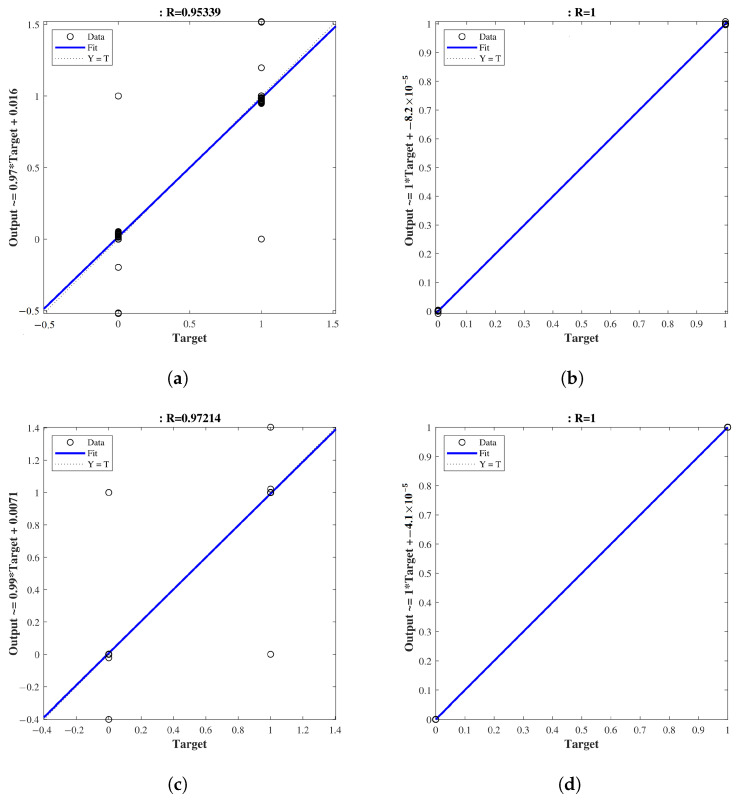
Scatter plots between the actual and predicted classes in the linear regression analysis of the model. In each configuration, the results show a strong correlation coefficient. (**a**) Orig-DB (10-CV). (**b**) Orig-DB (LOOCV). (**c**) Aug-DB (10-CV). (**d**) Aug-DB (LOOCV).

**Figure 10 healthcare-11-02675-f010:**
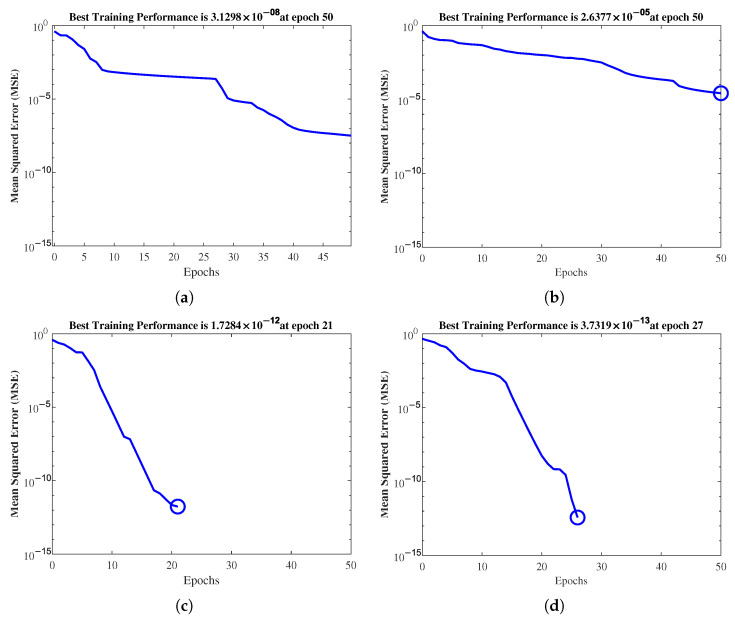
Model error curves in the training process in each configuration. The lower MSE when using the SMOTE-augmented database (Aug-db) in (**c**,**d**) suggests better-fitting models. (**a**) Orig-DB (10-CV). (**b**) Orig-DB (LOOCV). (**c**) Aug-DB (10-CV). (**d**) Aug-DB (LOOCV).

**Figure 11 healthcare-11-02675-f011:**
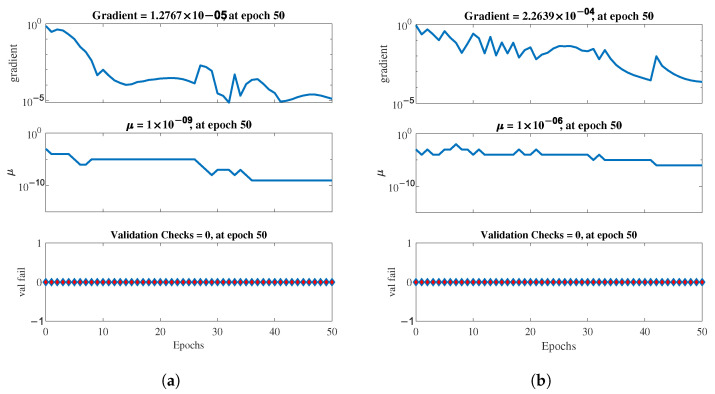
Plots representing a snapshot of the training state of our FFNN for each of the different configurations evaluated. (**a**) Orig-DB (10-CV). (**b**) Orig-DB (LOOCV). (**c**) Aug-DB (10-CV). (**d**) Aug-DB (LOOCV).

**Table 1 healthcare-11-02675-t001:** Concise demographic and smoking profile description of the 68 study participants.

Samples	Smoker			Age	Male	Total
	Yes	Former	No	μ±σ		
Control	8	13	20	57.34 ± 11.66	21 (51%)	41
OSCC	10	10	7	57.00 ± 13.87	19 (70%)	27
Total	18	23	27	57.20 ± 12.54	40 (59%)	68

**Table 2 healthcare-11-02675-t002:** Ranking of biomarkers using the *SVMAttributeEval* algorithm in the original database (Orig-DB) and SMOTE-augmented database (Aug-DB). The last column is the ranking difference between the Orig-DB and Aug-DB and shows the impact of synthetic samples by SMOTE on the ranking.

Rank	Orig-DB	Aug-DB (SMOTE)	Difference
1	X2.Hydroxyisovaleric.acid	Batyl.alcohol	−4
2	O.Phosphoethanolamine	Threonic.acid	−5
3	Batyl.alcohol	Maleic.acid	2
4	Threonic.acid	Threonine	2
5	Margaric.acid	X2.Hydroxyisovaleric.acid	−5
6	X2.Deoxy.glucose	X2.Deoxy.glucose	0
7	Threonine	O.Phosphoethanolamine	3
8	Adenine	Adenine	0
9	Tyramine	Oleic.acid	−6
10	Maleic.acid	Margaric.acid	7
11	Tryptamine	Phenylalanine	−20
12	Putrescine	Ornithine	−9
13	Norepinephrine	X2.Aminoethanol	−1
14	Hydroxylamine	Norepinephrine	−9
15	Dopamine	Tyramine	−35
16	Caproic.acid	Cadaverine	−30
17	Mannitol	Citramalic.acid	−7
18	X2.Aminoethanol	Palmitoleic.acid	5
19	Cadaverine	Sarcosine	3
20	Eicosapentaenoic.acid	X3.Aminopropanoic.acid	−9
21	Isoleucine	Putrescine	−13
22	Palmitoleic.acid	Elaidic.acid	4
23	Methionine.sulfone	Hydroxylamine	−5
24	Oleic.acid	Mannitol	15
25	Elaidic.acid	Glutamic.acid	3
26	Phenylalanine	Glycine	15
27	Citramalic.acid	Galactose	10
28	Sarcosine	Methionine.sulfone	9
29	X3.Aminopropanoic.acid	Eicosapentaenoic.acid	9
30	Arabitol	Inositol	−19
31	Glucose	Tryptamine	−4
32	Uridine	Mannose	−9
33	Mannose	Linoleic.acid	1
34	Galactosamine	Isoleucine	−4
35	Linoleic.acid	Glucose	2
36	X3.Aminoglutaric.acid	Palmitic.acid	−4
37	Ornithine	Ribitol	25
38	Ribitol	Galactosamine	1
39	Glycine	Acetoacetic.acid	13
40	Glutamic.acid	X3.Aminoglutaric.acid	15
41	Acetoacetic.acid	Uridine	2
42	Glucosamine	Octadecanol	−9
43	Glucono.1.5.lactone	Glucono.1.5.lactone	0
44	Galactose	Glucuronic.acid	17
45	meso.Erythritol	meso.Erythritol	0
46	Inositol	Caproic.acid	16
47	Glycolic.acid	Cysteine	−1
48	Glucuronic.acid	Glycolic.acid	4
49	Palmitic.acid	Arabitol	13
50	Cysteine	Dopamine	3
51	Octadecanol	Glucosamine	9

**Table 3 healthcare-11-02675-t003:** Overview of optimized parameters of our FFNN model.

Parameter	Value	Explanation
Input	From 1 to 51 features	Our biomarker database ranked by the *SVMAttributeEval* algorithm
Learning function	Levenberg–Marquardt	Used to update weight and bias values according to Levenberg–Marquardt optimization
Loss function	Mean-squared error	Measures model performance by calculating the mean-squared error between estimated values and actual values
Regularization rate	$1\times 10^{-5}$	For faster model convergence while preserving data representativeness
Number of neurons	3	Optimized with only 3 neurons in the hidden layer
Training process	10-CV/LOOCV	Cross-validation to improve model generalizability and estimate performance in practice
Transfer function	Tansig	Used in the hidden layer nodes to produce faster output rates
	Softmax	Used in the output layer nodes to provide better correlation coefficients for the processed hidden layer output data
Validation failures	5	Avoiding overfitting of the model

**Table 4 healthcare-11-02675-t004:** Comparative analysis of our OSCC diagnosis proposal with related literature studies.

Year	Study	Method	Eval.	Test Results (%)			
				ACC	TPR	TNR	AUC
2008	Hu et al. [18]	Logistic regression	LOOCV	-	90.0	83.0	93.0
2014	Wang et al. [62]	Logistic regression	LOOCV	-	92.3	91.7	87.1
2014	Yuvaraj et al. [15]	Statistical analysis	-	-	85.7	93.3	94.2
2016	Gleber-Netto et al. [63]	Fractional polynomial	-	-	-	-	87.2
2019	Deepthi et al. [64]	Statistical analysis	-	-	-	-	98.9
2020	Song et al. [12]	FS (Lasso regression)	20-CV	86.7	-	-	-
2020	Ishikawa et al. [42]	FS (logistic regression)	-	-	-	-	85.6
2021	Kouznetsova et al. [32]	FS (InfoGain) + NN	10-CV	79.5	-	-	-
2022	da Costa et al. [43]	FS (random forest)	10-CV	86.8	88.9	85.4	91.0
2022	Fonseca et al. [61]	FFNN multi-layer	10-CV	94.1	92.6	95.1	97.6
Our proposal							
2023	This work	FS (SVM) + FFNN	10-CV	98.5	96.3	100	96.5
2023	This work	FS (SVM) + FFNN	LOOCV	100	100	100	100
2023	This work	FS(SVM) + FFNN + DA	10-CV	98.8	100.0	97.6	99.2
2023	This work	FS(SVM) + FFNN + DA	LOOCV	100.0	100.0	100.0	100.0

Eval.: classification evaluation scheme, ACC: accuracy, TPR: sensitivity, TNR: specificity, AUC: area under the ROC curve, LOOCV: leave-one-out cross-validation, *k*-CV: *k*-fold cross-validation, NN: neural network, FFNN: feed-forward neural network, FS: feature-selection process, DA data augmentation by SMOTE.

## Data Availability

The dataset used in this study was the same as that available in [43].

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
