# Peer review of "An Intelligent System to Improve Diagnostic Support for Oral Squamous Cell Carcinoma"

_healthcare, 2023, doi:10.3390/healthcare11192675_

Round 1

Reviewer 1 Report

Raise a fundamental question: is oral carcinoma difficult to find?

If there is possibility of having to a carcinoma, going to the doctor is recommend. Because, it is extremely easy to take probe from oral cavity.

The authors should discuss possible scenarios of situational development with this method for above question.

Author Response

Raise a fundamental question: is oral carcinoma difficult to find?

If there is possibility of having to a carcinoma, going to the doctor is recommend. Because, it is extremely easy to take probe from oral cavity.

The authors should discuss possible scenarios of situational development with this method for above question.

OSCC is hard to diagnose in its early stages, especially because patients are not aware of symptoms and signs, which mimic regular oral irritations, ulcerations, leukoplakia and gingivitis. So, small injuries can be misinterpreted, allowing the cancer to evolve. Classical OSCC diagnosis relies on a biopsy that is conducted, required only after clinical observation; however, injuries are already visible and likely too advanced. Therefore, a simple process for collecting salivary samples to gather biomarkers is not only quick, cheaper, and non-invasive but also can become a general clinical protocol for prevention. Finally, analyzing the biomarkers is a hard task, so our work aims to evaluate their impact on classifiers for early diagnosis of OSCC. 

We have added this to our manuscript, to clarify to the reader. We appreciate your suggestion.

Furthermore, the whole manuscript has been revised for clarity issues, including images quality and captions.

Reviewer 2 Report

This is a well-designed study and well-written manuscript. I have a couple of comments as below.

When building prediction models using relatively small numbers of samples, general concerns are the possibility of overfitting and the difficulty of extrapolating the model when applying to new samples. Even though cross validation has been performed, but with both model building and variable selection (the best model from the subsets of predictors) based on the same set of data, the re-use of data can potentially lead to overfitting.

The difficulty of extrapolating the model when applying to new samples arises when the samples used for training the model are too few, and are not representative of a wider population.   

Other minor comments:  

In Figure 5, it looks like adding more features for prediction sometimes resulted in severe reduction in prediction performance. For example, in figure 5a, the accuracy dropped from 98.53% to ~75% when numbers of predictors increased from 14 to 15.

How do we intepret these drastic changes in performance?   

Some method details might be helpful in a few places. For example,   -  It looks like data normalization had been applied to the biomarkers before calculating the Euclidean distances for Figure 2 and showing the distributions in Figure 3.

How was the data normalized?   - For the SMOTE data augmentation, "k nearest minority class neighbors" were used.

What was the choice of `k` used in this analysis?

Author Response

This is a well-designed study and well-written manuscript. I have a couple of comments as below.

We are very glad for your time reviewing our work, your suggestions and complements. We worked very hard on this project.

When building prediction models using relatively small numbers of samples, general concerns are the possibility of overfitting and the difficulty of extrapolating the model when applying to new samples. Even though cross-validation has been performed, but with both model building and variable selection (the best model from the subsets of predictors) based on the same set of data, the re-use of data can potentially lead to overfitting.

The difficulty of extrapolating the model when applying to new samples arises when the samples used for training the model are too few, and are not representative of a wider population.    

Other minor comments:  

In Figure 5, it looks like adding more features for prediction sometimes resulted in severe reduction in prediction performance. For example, in figure 5a, the accuracy dropped from 98.53% to ~75% when numbers of predictors increased from 14 to 15.

How do we intepret these drastic changes in performance?   

We understand your concern and we appreciate your comment. In fact, adding a single feature from 14 to 15 presented a drop in terms of accuracy. However, that is caused by the algorithm SVMAttributeEval behavior that ranks each class' features separately using a one-vs-all (OvA) strategy and then "negotiating" from the top of each stack to get a final ranking.  So, a small increase in the number of features impacts the model accuracy. On the other hand, SVMAttributeEval helps in finding the appropriate number of features to maximize accuracy. We have included in our manuscript that explanation to help a reader understand the process. Moreover, we changed Figures 5 and 6 to start the Y-axis from zero and avoid scaling problems.

Some method details might be helpful in a few places. For example,   -  It looks like data normalization had been applied to the biomarkers before calculating the Euclidean distances for Figure 2 and showing the distributions in Figure 3. How was the data normalized?   - For the SMOTE data augmentation, "k nearest minority class neighbors" were used.

We are glad you brought that to our attention. Our data was normalized using Z-score normalization. We added this information to our manuscript.

What was the choice of `k` used in this analysis?

We used a k=5, which was defined empirically, considering the number of original samples available (27), in order to balance the maintenance of important information and better sample diversity. Thus, we seek to improve the generalization of the model, avoiding both overfitting and underfitting. We added this comment to our text. Thanks for bringing this to our attention.

Reviewer 3 Report

Dear Editor,

I would like to express my deep thanks to you for allowing me to review this valuable manuscript, " An intelligent system to improve diagnostic support for oral squamous cell carcinoma". The objective of this study is to develop a technique for assisting in the diagnosis of Oral Squamous Cell Carcinoma (OSCC) by using a shallow artificial neural network. The primary purpose is to differentiate between individuals with OSCC and healthy individuals based on salivary biomarker data.

The paper needs the following changes:

1.      The authors need to expand on the pathophysiology of oral squamous cell carcinoma in the introduction section

2.      In the introduction section, Oral Squamous Cell Carcinoma (OSCC) is repeated. Please reconsider your use of abbreviations and define them when they are first used in the text only.

3.      Some abbreviations need to be defined in the “Related Work” section as Q-TOF liquid chromatography

4.      The whole manuscript needs careful proofreading for minor spacing, syntax, and language corrections in some places.

5.      A list of abbreviations should be added to the manuscript

6.      Figures and Tables must be self-explanatory; thus, abbreviations must be defined in footnotes so that readers may examine them without having to go to the main text

7.      The authors created beautiful figures, but figures 6 & 10 in the manuscript are not clear enough. The original or high-definition figures should be provided

Author Response

Dear Editor,

I would like to express my deep thanks to you for allowing me to review this valuable manuscript, " An intelligent system to improve diagnostic support for oral squamous cell carcinoma". The objective of this study is to develop a technique for assisting in the diagnosis of Oral Squamous Cell Carcinoma (OSCC) by using a shallow artificial neural network. The primary purpose is to differentiate between individuals with OSCC and healthy individuals based on salivary biomarker data.

We are very glad the time and effort you spent reading and commenting your work. Thanks a lot for the compliments.

The paper needs the following changes:

1. The authors need to expand on the pathophysiology of oral squamous cell carcinoma in the introduction section

We have expanded our introduction to provide a much clearer explanation of the pathophysiology of oral squamous cell carcinoma. We appreciate your suggestion.

2. In the introduction section, Oral Squamous Cell Carcinoma (OSCC) is repeated. Please reconsider your use of abbreviations and define them when they are first used in the text only.

Thank you for bringing this to our attention. We fixed it in the manuscript.

3. Some abbreviations need to be defined in the “Related Work” section as Q-TOF liquid chromatography

We defined all abbreviations in the Related Work section, and we did our best to make sure in the whole manuscript abbreviations are defined before first use.

4. The whole manuscript needs careful proofreading for minor spacing, syntax, and language corrections in some places.

We performed a language checking to improve our text.

5. A list of abbreviations should be added to the manuscript

We added a list of abbreviations to the end of the manuscript.

6. Figures and Tables must be self-explanatory; thus, abbreviations must be defined in footnotes so that readers may examine them without having to go to the main text

We appreciate the comment. We have added more explanatory comments to all abbreviations; however, terms already introduced were kept to avoid repetition. 

7. The authors created beautiful figures but figures 6 & 10 in the manuscript are not clear enough. The original or high-definition figures should be provided

Thanks for your comment. We have adjusted all images in our manuscript, not only in terms of resolution, but we also put all of them in the same scale to provide a more fair comparison. In any event, all images are provided in EPS, which is a vectorial format, so that the publishing desk can change the size of the image with no quality loss and no aspect ratio change.

Round 2

Reviewer 1 Report

Thank you for your good revised.